# Purple Urine Bag Syndrome in a Home-Dwelling Elderly Female with Lumbar Compression Fracture: A Case Report

**DOI:** 10.3390/healthcare11162251

**Published:** 2023-08-10

**Authors:** Milka B. Popović, Deana D. Medić, Radmila S. Velicki, Aleksandra I. Jovanović Galović

**Affiliations:** 1Department of Hygiene, Faculty of Medicine, University of Novi Sad, Hajduk Veljkova 3, 21000 Novi Sad, Serbia; radmila.velicki@mf.uns.ac.rs; 2Center for Hygiene and Human Ecology, Institute of Public Health of Vojvodina, Futoška 121, 21000 Novi Sad, Serbia; 3Department of Microbiology, Faculty of Medicine, University of Novi Sad, Hajduk Veljkova 3, 21000 Novi Sad, Serbia; deana.medic@mf.uns.ac.rs; 4Center for Microbiology, Institute of Public Health of Vojvodina, Futoška 121, 21000 Novi Sad, Serbia; 5Faculty of Pharmacy Novi Sad, University of Business Academy, Trg Mladenaca 5, 21000 Novi Sad, Serbia

**Keywords:** purple urine bag syndrome, purple urine discoloration, urinary tract infection, *Proteus mirabilis*, chronic indwelling urinary catheter, spinal cord injury, case report

## Abstract

Purple urine bag syndrome (PUBS) is an uncommon, but usually benign, underrecognized clinical condition with the distressing presentation of purple, blue or reddish discoloration of a patient’s catheter bag and tubing in the setting of catheter-associated urinary tract infections (UTIs). PUBS is the result of the complex metabolic pathway of the dietary essential amino acid tryptophan. Its urinary metabolite, indoxyl sulfate, is converted into red and blue byproducts (indirubin and indigo) in the presence of the bacterial enzymes indoxyl sulfatase and phosphatase. The typical predisposing factors are numerous and include the following: female gender, advanced age, long-term catheterization and immobilization, constipation, institutionalization, dementia, increased dietary intake of tryptophan, chronic kidney disease, alkaline urine, and spinal cord injury (SCI). Here, we present a case of PUBS in a home-dwelling elderly female patient with a history of long-term immobility after a pathological spinal fracture, long-term catheterization, constipation, and malignant disease in remission. Urine culture was positive for *Proteus mirabilis*. This state can be alarming to both patients and physicians, even if the patient is asymptomatic. Healthcare professionals and caregivers need to be aware of this unusual syndrome as an indicator of bacteriuria in order to initiate proper diagnostics and treatment.

## 1. Introduction

Purple urine bag syndrome (PUBS) is an uncommon, but usually benign, clinical condition with the purple discoloration of the patient’s catheter bag and tubing [1]. The characteristic purple color is the result of the presence of predominantly Gram-negative bacteria in the urine. This urinary tract infection (UTI) may be completely asymptomatic or present with the classical bacteriuria symptoms. The mechanism of the characteristic discoloration of the Foley bag and the urine involves specific microbial enzymes. Bacteria-derived indoxyl sulfatase and phosphatase cause blue (indigo)- and red (indirubin)-pigmented breakdown products of dietary amino acid tryptophan metabolite (indoxyl sulfate). The different proportions of these two pigmented metabolites produce various shades of discoloration, ranging from purple and blue to a reddish color (Figure 1) [2,3]. The formed pigments predominantly bind to the synthetic materials, such as tubing and bags, whereas the fresh urine itself is not discolored.

According to systematic reviews, typical predisposing factors include female gender, advanced age, long-term immobilization and catheterization, constipation, institutionalization, dementia, increased dietary intake of tryptophan, chronic kidney disease, alkaline urine, and spinal cord injury (SCI).

In the study conducted by Sabanis et al., the mean age of 246 affected PUBS patients was 78.9 ± 12.3 years, of which 70.7% were female, with a high prevalence of alkaline urine (91.3%) and constipation (90.1%). More than 76% were bed-ridden, 45.1% were experiencing long-term catheterization, 42.8% had been diagnosed with dementia, 14.3% had recurrent urinary tract infections, and 14.1% were chronic kidney disease (CKD) patients [4].

The most common microbes identified as the causes of UTIs related to PUBS were as follows: *Escherichia coli*, *Proteus mirabilis*, *Klebsiella pneumoniae*, *Enterococcus*, *Pseudomonas aeruginosa*, *Providencia stuartii*, *Providencia rettgeri*, *Morganella morgannii*, *Enterococcus faecalis*, etc. [4,5].

## 2. Case Presentation

Our case was a 79-year-old female with a history of long-term immobility after pathological spinal compression fracture, and urinary retention managed with a chronic indwelling Foley catheter, recurrent asymptomatic urinary tract infections, constipation, hypertension, prior malignant disease in remission, osteoporosis, and periodically altered mental status. The patient’s routine medications included paracetamol, pantoprazol, diltiazem, ramipril, diazepam, vitamin D supplements, and melperon hydrochlorid.

The family reported that the patient lived at home with a professional home care service during the day. A compression fracture of the lumbar (lower) spine and a neurogenic bladder were diagnosed one year prior to the first urine discoloration. The first purple urine bag discoloration was misinterpreted as a dietary-related urine discoloration associated with beetroot or rosehip tea consumption, and did not receive an appropriate assessment or treatment. At the moment of the first PUBS onset, the catheter was 15 days “old”. The family and caregiver noticed a recurrent light-purple discoloration of the patient’s urine bag and a strong urine odor. The patient did not complain of any symptoms that could indicate urinary tract infection. In the following days, the family noticed a change in mental status indicated by mild signs of mental confusion.

The patient’s overall clinical presentation was unremarkable. Apart from the elevated serum levels of lactic acid dehydrogenase (LDH), most biochemical parameters did not show any significant changes. Slightly elevated creatinine levels (reference interval: 49–97 μmol/L; the determined value was 115 μmol/L) and liver enzyme activity did not raise any concerns at the time. The physical examination recorded a slightly confused elderly female in no acute distress, with dry mucous membranes, active bowel sounds, without tenderness to deep low abdominal palpation, with both-sided lower extremity weakness and paresis, muscular atrophy, and a Foley bag with purple urine, as shown in Figure 2.

The purple color of the PE urine bag with the extended tubing was more intense than the color of the silicone catheter. A clear demarcation zone between the two grades of the same purple color was at the level of the urine drainage port of the Foley catheter. The more intense purple color of the urine tube implied that biofilm had formed.

Urinalysis revealed brownish, cloudy urine positive for nitrites, with more than 100 white blood cells per high-power microscopy field, without hematuria.

Immediately after the Foley catheter and urine bag change, the caregiver started with empiric antibiotics (fosfomycin), intensive per-oral rehydration, and bowel function normalization. Urine color and urine odor were normalized within 24 h. When the urine culture results became available, the patient started an antibiotic treatment course according to culture sensitivities. The urine color returned to clear yellow, without any sediment or purple shade. This case report was composed according to the CARE guidelines [6].

### Microbiological Examination

A urine sample was collected from a urinary catheter and the complete urine bag was delivered to the laboratory. Urine samples were seeded with a sterile, calibrated, plastic 1 µL inoculation loop on the chromogenic medium CHROMID^®^CPS^®^ Elite agar (bioMérieux, Marcy-l’Étoile, France). After incubation for 18–24 h at 35–37 °C, the plates were examined and a significant number of colonies of >10^5^ CFU/mL of yellow-brown color with a characteristic smell of ammonia were established. MALDI-TOF/MS (Bruker, MA, USA) was used as an analytical tool for a further determination of the bacterial species in the urine samples.

The urine culture from both the sterile vial and Foley bag was positive for *Proteus mirabilis* (*P. mirabilis*) (Figure 3).

After identification, an antibiotic sensitivity test was performed using the Kirby–Bauer disc diffusion method on Mueller–Hinton agar according to the recommendations of the European Committee on Antimicrobial Susceptibility Testing (EUCAST) standard [7]. The isolates showed sensitivity to aminopenicillins, aminopenicillins with beta-lactamase inhibitors, cephalosporins, aminoglycosides, quinolones, cotrimoxazole, and carbapenems (meropenem).

## 3. Discussion

The purple urine bag phenomenon was first described by Barlow and Dickson in 1978, who described the rare mechanism of purple color formation in urine bags in children with spina bifida in pediatric wards. The investigation elucidated the bacterial decomposition of dietary amino acid tryptophan in the gut lumen, producing indoxyl sulfate, which is consequently excreted in the urine, then oxidized to insoluble indigo upon contact with the air in a collecting bag [8,9]. A similar clinical manifestation in infants, known as “blue diaper syndrome” (Drummond syndrome), is also a consequence of abnormal tryptophan metabolism but is, however, genetically caused. In adults, a wide array of conditions may produce urinary discoloration: hematuria, myoglobinuria, porphyria, alkaptonuria, lipiduria, and the presence of certain microorganisms. In addition, numerous medications (amitriptyline, ibuprofen, propofol, L-dopa, phenytoin, flutamide, senna, and laxatives with a phenolphthalein component) as well as the consumption of intensively colored foods (beetroots, blackberries, fava beans, carrots, and rosehip tea) may produce changes in urine color, ranging from red or orange to blue-green [10,11].

Since it cannot be synthesized in the human body, tryptophan—an amino acid with an indole nucleus—is one of the essential amino acids and must be obtained through the diet. Tryptophan-rich foods include oats, bananas, dried prunes, milk, tuna fish, cheese, bread, chicken, turkey, peanuts, and chocolate [12]. In the gastrointestinal (GI) tract, due to the action of the GI microbiome, several metabolites of tryptophan can be produced. Most of these compounds arise from proteins which are not fully digested and, thus, are malabsorbed and, hence, remain in the colon (Figure 4). Bacteria may give rise to a number of tryptophan-derived, metabolically active molecules. It was experimentally proven that the amount of indole produced by bacteria is proportional to the amount of supplied tryptophan [13]. So, it is clear that tryptophan in the form of proteins incompletely digested from the human diet is a substrate for the intestinal microbiota. One of the enzymes responsible for indole production in the colon is tryptophanase [14,15].

In the case presented herein, the most probable dietary sources of tryptophan were bananas, meat, and chicken pâté, which were consumed in excessive amounts for a longer period before the onset of the urinary discoloration.

Indole is a precursor for several pivotal mediators including tryptamine, serotonin, melatonin, kynurenines, and nicotinic acid. Other indoles are considered waste products and are often conjugated prior to urinary excretion. The majority of indole is oxidized to indoxyl and conjugated with sulfate in the liver [16].

It has also been suggested that indoxyl sulfate is toxic to renal tubular cells, while increased indoxyl sulfate levels accelerate the progression of renal disease [17]. Thus, the produced physiologically active metabolites may be linked to the patient’s altered mental state during bacteriuria.

The 3D structure of urinary sediments and urine bag walls of two cases of PUBS were observed using Low-Vacuum Scanning Electron Microscopy (LVSEM), which showed granular purple crystals around the bacilli, cocci, or mycelium that adhered to the walls of the bag [18].

As mentioned before, the discoloration in PUBS occurs when the two pigments come in contact with the synthetic materials of the catheter tubing and urinary bag, presumably already covered with biofilm. In some cases, the whole urine bag and the catheter are discolored. In other cases, the level of discoloration clearly indicates the area of prolonged contact between the urine and the synthetic material. A frequent misinterpretation is that the urine in PUBS itself is discolored [19,20].

*P. mirabilis* is a Gram-negative rod-shaped bacterium most noted for its swimming and swarming motility phenomenon and urease activity. It frequently causes catheter-associated urinary tract infections that are often polymicrobial, particularly in patients undergoing long-term catheterization. These infections may be accompanied by the development of bladder (urolithiasis) or kidney stones due to the alkalinization of urine from urease-catalyzed urea hydrolysis [21].

*P. mirabilis* is an opportunistic pathogen that uses a diverse set of virulence factors to access and colonize the host urinary tract and develop a crystalline biofilm [22]. Biofilm formation is a multistep process, which begins with migration by swarming motility along the catheter, followed by attachment to the surface through fimbriae adhesins. Once firmly attached, the number of *P. mirabilis* cells rises. They produce a substantial amount of urease enzyme which hydrolyses urea present in urine to ammonia and bicarbonate anion, inevitably raising the pH of the surrounding environment and leading to mineral precipitation in crystal form [22,23]. The increase in alkalinity leads to the deposition of struvite (ammonium magnesium phosphate) and hydroxyapatite (calcium phosphate) crystals on a developing biofilm. This process, named ureolytic biomineralization, is additionally aided by *P. mirabilis* capsular polysaccharide for biofilm matrix formulation and peptide efflux [23,24]. Fully developed crystalline biofilm formation by *P. mirabilis* unavoidably causes frequent catheter encrustation and blockage and, in most cases, is accompanied by urine retention and ascending UTIs, which may even lead to antibiotic treatment resistance [25,26].

*P. mirabilis* can be found in a wide variety of environments, including soil, water sources, and sewage, but it is predominantly a commensal of the gastrointestinal tracts of humans and animals. The majority of *P. mirabilis* UTIs are a consequence of the ascension of bacteria from the GI tract. There is literature evidence showing that patients with *P. mirabilis* infection have exactly the same bacterial strain present both in urine and the stool samples [22]. A certain percentage of cases are the result of person-to-person transmission, frequently in hospitals or nursing homes.

Several bacterial species that commonly infect the bladder after spinal cord injury can hydrolyze urea; apart from *Proteus,* these species are as follows: *Pseudomonas*, *Providencia*, *Klebsiella*, and *Morganella*. Interestingly, the *Proteus*’ urease is one of the fastest-acting enzymes, so it is not surprising that this microbe is most commonly involved in catheter encrustation and blocking. Moreover, it may also lead to the formation of stones in the urinary tract, particularly in indwelling patients. Once stones are formed, antimicrobial therapy is usually not effective in eradicating *Proteus* and other organisms lodged in the of stones’ crevices [27,28,29]. These infections are common in long-term catheterized patients, such as those who reside in nursing homes and chronic care facilities, and may be of particular danger to spinal cord injury patients. Recurrent UTIs are often seen in patients with spinal cord injuries (SCIs) and neurogenic bladders. It is well known that impaired bladder emptying, repeated catheterization, vesicoureteral reflux, and similar conditions that impair normal urine excretion are predisposing factors for infections of the urinary tract. Immobility is also one of the factors to be considered. In SCI patients, *Proteus* is more frequently found in individuals with a higher degree of damage to the spinal cord. In those patients, hospitalization, the onset of decubitus, and the use of indwelling catheter are highly probable. *Proteus* detection in urine is one of the predictive factors for urologic complications in persons with SCI [27,30]. Results from our laboratory for the previous period (2020–2023) show an average incidence of *P. mirabilis* presence in urine samples of 1.91% for both hospitalized patients and outpatients in the region of Vojvodina, Serbia (Table 1).

Generally speaking, the incidence of PUBS in catheterized patients is fairly low, although the percentages vary depending on the geographical region and the population group enrolled in the study. According to the meta-analysis by Llenas-García [31], the prevalence of PUBS in observational studies was 11.7% in patients with long-term urinary catheterization. It is a widely accepted notion that PUBS may only reflect asymptomatic bacteriuria and, therefore, should not be treated with aggressive antibiotic therapy. However, taking into account data obtained in vitro [32] on the immunomodulatory capacity of *Proteus mirabilis* strains, it is worth considering the possibility of the scenario where *P. mirabilis* infection subdues the host immune response and that is the reason for the lack of symptoms in patients. If left untreated, such an infection may lead to a propagation of the possibility of biofilm formation on the catheter surface (possibly also on the bladder epithelium). Once formed, biofilm represents “a refuge” for microorganisms, protecting them from antibiotic action and the host immune cells. Well-established biofilm further leads to complications due to changes in the microenvironment (ammonia production, pH elevation, and crystal formation). While elevated ammonia concentrations are toxic on their own, crystals lead to catheter encrustation and blockage and are recorded as constituents of kidney stones in such patients [33,34].

Biofilm formation inhibition on catheters could become a promising alternative to conventional antimicrobial-based treatment, which may be associated with rapid resistance development [35]. Various modifications of materials and antimicrobial coatings on the catheter surface surely offer one of the possible solutions in order to prevent PUBS. The use of hydrogels, polytetrafluoroethylene (PTFE), polyethylene glycol (PEG), polyzwitterions, and specific enzymes as coatings all involve material modifications. As antimicrobial coatings, apart from various antibiotics (chosen for specific infecting agents), metallic ions, nanoparticles, nitric oxide bacteriophages, and antimicrobial peptides (AMPs) may be used in order to prevent biofilm formation [36]. Although there are numerous data from in vivo, in vitro, and clinical studies, a unanimous conclusion on which coating is the right choice for a specific pathogen has not been reached. Furthermore, with the possibilities offered by 3D printing, new options are available, like the coating of multiple drugs on catheters with different release profiles. Taking into account the fact that several phytochemicals—like curcumin, allicin, or proanthocyanidins—have shown in vitro effectiveness against *P. mirabilis* biofilm formation [23,24], it seems that there are a plethora of options available. According to some authors, a combination of various approaches will give the best results. However, more research efforts are needed in order to effectively translate scientific findings into clinical practice.

## 4. Conclusions

In this case report, we confirm that female gender, increased dietary tryptophan, long-term immobilization and catheterization, severe constipation, high urinary bacterial load, and renal failure are the risk factors for purple urine bag syndrome. Although this condition seems benign, it sometimes requires aggressive management.

In addition, we argue that purple urine bag syndrome should not be taken lightly, even if the patient is asymptomatic. In our opinion, it should be treated with the notion that it is a “warning signal”. It indicates an ongoing urinary tract infection, which could develop into a serious health threat to the patient, usually heavily burdened with other chronic conditions in already-vulnerable patients [37].

Healthcare professionals—particularly urologists and geriatricians—need to be aware of this unusual syndrome so that an appropriate investigation and possible treatment can be initiated as soon as possible [38]. In spite of the fact that PUBS is a fairly rare condition today, with an aging population worldwide, its incidence is likely to increase in the future. In that respect, innovative solutions for biofilm prevention, combined with an effective antimicrobial therapy, may be of relevance.

## Figures and Tables

**Figure 1 healthcare-11-02251-f001:**
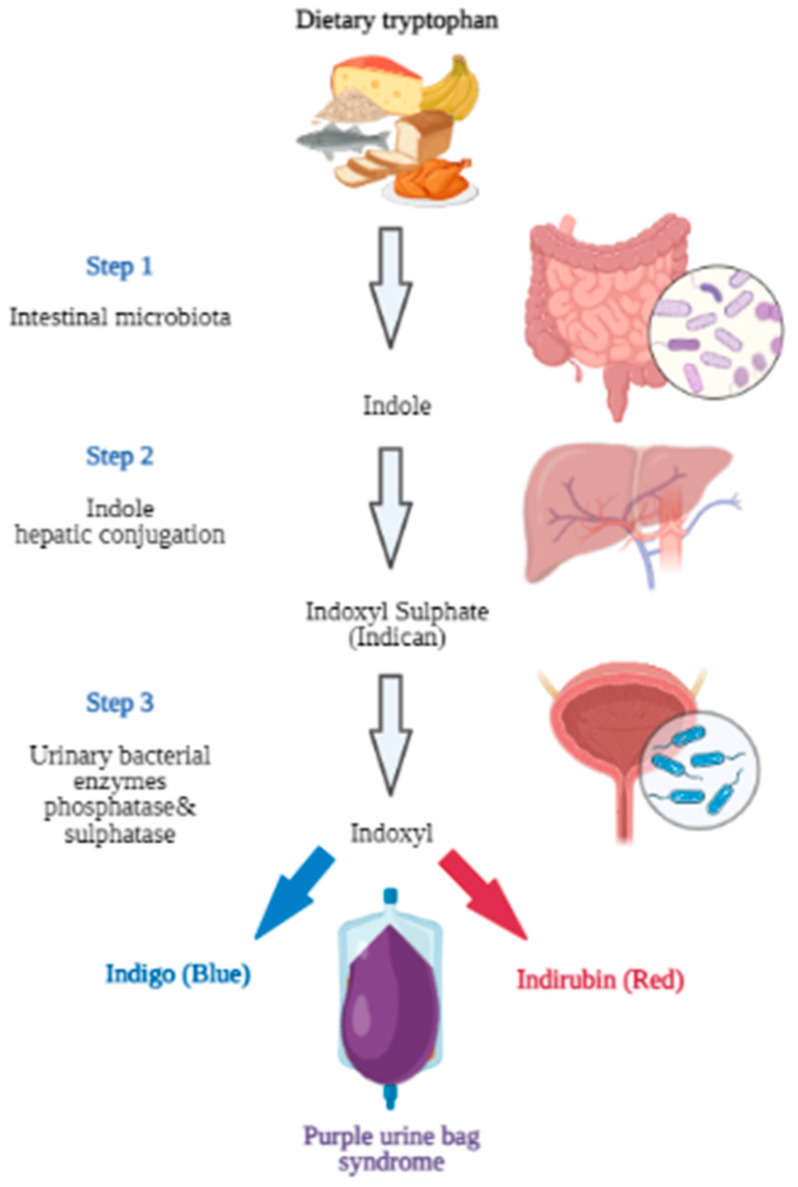
Metabolic pathway of dietary tryptophan leading to purple urine bag syndrome (PUBS) (created with BioRender.com).

**Figure 2 healthcare-11-02251-f002:**
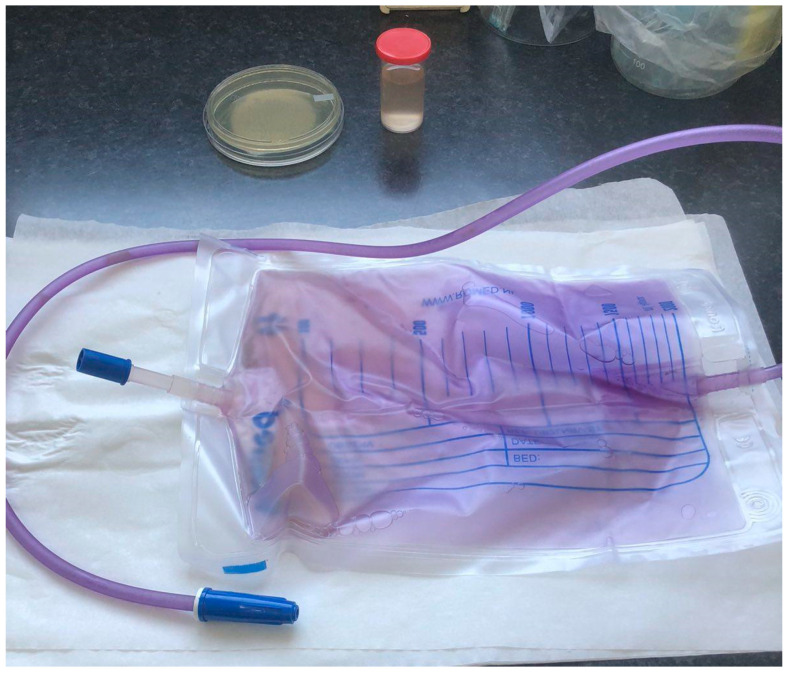
The purple urine bag and yellow-brown color of voided urine.

**Figure 3 healthcare-11-02251-f003:**
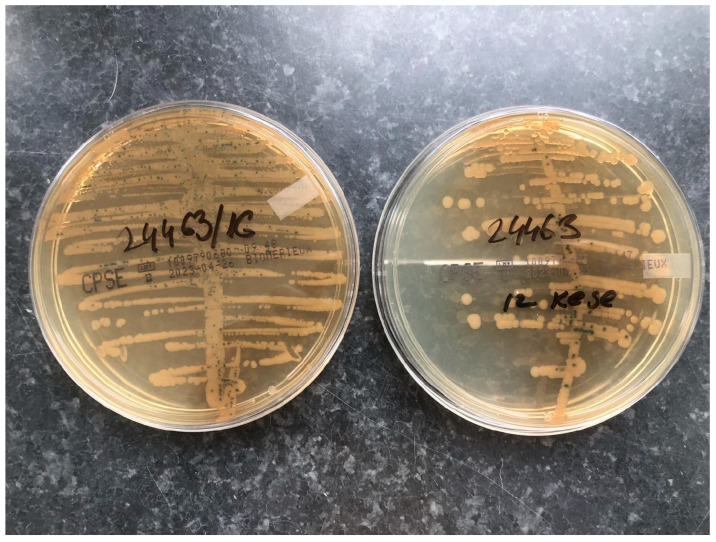
A positive urine culture on CHROMID^®^CPS^®^ Elite agar (**left**—sample from the vial; **right**—sample from Foley bag).

**Figure 4 healthcare-11-02251-f004:**
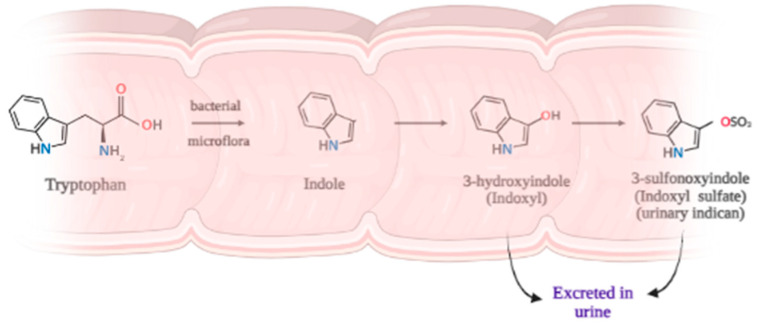
Pathway of bacterial transformation of tryptophan in the colon (created with BioRender.com).

**Table 1 healthcare-11-02251-t001:** Overall number of *P. mirabilis* positive urine analysis and the incidence (%) in the years 2020–2023 (until 30 June). Samples were obtained from both hospitalized patients and outpatients in the Vojvodina region (Serbia).

Year	Number of Microbiological Urine Analyses ^1^	Number of *P. mirabilis* Detected	Incidence %
2023	21,558	410	1.90
2022	41,544	830	2.00
2021	34,730	658	1.89
2020	27,049	490	1.81
Total	124,881	2388	1.91 ^2^

^1^ All analyses were performed at the Center for Microbiology, Institute of Public Health of Vojvodina, Serbia; ^2^ average incidence.

## Data Availability

The data presented in this study are available on request from the corresponding author. The data are not publicly available due to the patient’s privacy protection.

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
