# Peer review of "Purple Urine Bag Syndrome in a Home-Dwelling Elderly Female with Lumbar Compression Fracture: A Case Report"

_healthcare, 2023, doi:10.3390/healthcare11162251_

Round 1

Reviewer 1 Report

The manuscript is well written.  I only recommend that in each specific case the authors indicate what was the established or suspected source of the dietary tryptophan that was included in the manuscript's discussion on page 4.

Author Response

Dear Reviewer,

thank you for your comments and suggestions. We have tried to make ammendments accordingly in order to improve the quality of our manuscript.

Here are our responses for each of your comments:

The manuscript is well written.  I only recommend that in each specific case the authors indicate what was the established or suspected source of the dietary tryptophan that was included in the manuscript's discussion on page 4.

“In our case, the most probable dietary sources of tryptophan were bananas, meat and chicken pâté, which were consumed in excessive amounts for a longer period before the onset of the urinary discoloration.”

Correction made

Sincerely,

the authors

Reviewer 2 Report

In the case reported there is the focusing on the physical examination and urinalysis. There is not a mention about blood findings. Could be insteresting to know if there were some changes to the reference value to complete the case presentation. 

The quality of english is good. The whole paper is easly understandble.

Author Response

Reviewer 2

Dear Reviewer,

thank you for your comments and suggestions. We have tried to make ammendments accordingly in order to improve the quality of our manuscript.

Here are our responses for each of your comments:

In the case reported there is the focusing on the physical examination and urinalysis. There is not a mention about blood findings. Could be insteresting to know if there were some changes to the reference value to complete the case presentation. 

“The patient’s overall clinical presentation was unremarkable. Apart from the elevated serum levels of lactic acid dehydrogenase (LDH), most biochemical parameters did not show any significant changes. Slightly elevated creatinine levels (reference interval 49-97 μmol/L, determined value was 115 μmol/L) and liver enzyme activity, did not raise any concerns at the time. “

Correction made

Sincerely,

the authors

Reviewer 3 Report

Dear Authors,

I have carefully reviewed your case report and would like to provide constructive feedback on the manuscript. Overall, I find the topic of the case report to be captivating and highly relevant, offering valuable insights to the readers. I commend the well-structured introduction and case presentation, which effectively lay the foundation for understanding the subject matter. 

However, I suggest considering an alternative title for the "Urine and antibiogram" section. Perheps "Microbiological examination" would be more more fitting and accurate title. 

While the discussion section is well-written, I believe it would greatly benefit from the inclusion of a discussion on the considered differential diagnoses and the rationale behind their exclusion. This additional analysis would enhance the comprehensive nature of the case report.

I must commend you on the inclusion of relevant and easily understandable images and figures, as they significantly enhance the presentation and explanation of the case.

Furthermore, I appreciate the selection of relevant and up-to-date references that directly contribute to the validity and currency of the described case report.

However, I would like to draw your attention to the need for improvement in the language and writing style of the manuscript. Enhancing the clarity and overall impact of the text through refinement would further strengthen the quality of your work.

In conclusion, with the suggested revisions and improvements, I believe your case report has the potential to make an even greater contribution to the scientific community.

Thank you for your valuable contribution to the field.

There is room for improvement in the language and writing style of the manuscript.

Author Response

(The authors gave the same response as above.)

Reviewer 4 Report

In the manuscript, the authors present a case report of interesting purple urine syndrome one of the patients in a good way. Although this syndrome has been known and described for a long time, due to the fact that such cases are rare and difficult for some diagnosticians and doctors to interpret, it is worth mentioning them.Below are some of my remarks and comments to the text of the article:

- the authors in the text indicate that this syndrome is uncommon. What exactly is its frequency of occurrence?

- in the text there is also information that earlier in this patient (about two weeks) there was purple urine, was Proteus mirabilis also isolated in that case, so it will be a recurrent infection with the same pathogen?

-  in the discussion stated " A biofilm formation inhibition on catheters could become a promising alternative to conventional antimicrobial-based treatment that is associated with rapid resistance development” please provide more details or an example of such a procedure

- please pay attention to the spelling of the abbreviation UTI or CAUTI. On page 8 there should be eg CAUTI's

Author Response

Reviewer 4

Dear Reviewer,

thank you for your comments and suggestions. We have tried to make amendments accordingly in order to improve the quality of our manuscript.

Here are our responses for each of your comments:

In the manuscript, the authors present a case report of interesting purple urine syndrome one of the patients in a good way. Although this syndrome has been known and described for a long time, due to the fact that such cases are rare and difficult for some diagnosticians and doctors to interpret, it is worth mentioning them. Below are some of my remarks and comments to the text of the article:

- the authors in the text indicate that this syndrome is uncommon. What exactly is its frequency of occurrence?

“Results from our laboratory for the previous period (2020-2023) show average incidence of P. mirabilis detection in urine samples of 1.91% for both hospitalized patients and outpatients in the region of Vojvodina, Serbia (Table 1.).

Generally speaking, the incidence of PUBS in the population is low, although the percentages vary depending on the geographical region and population group. According to the meta-analysis of Llenas-García, the prevalence of PUBS in observational studies was 11.7% in patients with long-term urinary catheterization.”

Corrections made

- in the text there is also information that earlier in this patient (about two weeks) there was purple urine, was Proteus mirabilis also isolated in that case, so it will be a recurrent infection with the same pathogen?

As we mentioned in the text, the first purple urine bag discoloration was misinterpreted as dietary-related urine discoloration related to beetroot or rosehip tea consumption and did not receive appropriate assessment and treatment.

Since the patient was cared for in home-dwelling setting, at that point, microbiological analysis of urine was not performed.

-  in the discussion stated " A biofilm formation inhibition on catheters could become a promising alternative to conventional antimicrobial-based treatment that is associated with rapid resistance development” please provide more details or an example of such a procedure

“Various modifications of materials and antimicrobial coatings on the catheter surface surely offer one of the possible solutions in order to prevent PUBS. Use of hydrogels, Poly(Tetrafluoroethylene) (PTFE), Polyethylene Glycol (PEG), polyzwitterions, and specific enzymes as coatings – all involve material modifications. As antimicrobial coatings, apart from various antibiotics (chosen for specific infecting agents), metallic ions, nanoparticles, nitric oxide bacteriophages and antimicrobial peptides (AMPs), may be used in order to prevent biofilm formation (Kanti et al., 2022). Although there are numerous data from in vivo, in vitro and clinical studies, a unanimous conclusion on which coating is the right choice for a specific pathogen, has not been reached. Furthermore, with the possibilities offered by 3D printing new options are available, like coating of multiple drugs on catheters with different release profiles. Taking into account the fact that several phytochemicals – like curcumin, allicin or proanthocyanidins- have shown in vitro effectiveness against P. mirabilis biofilm formation [21], it seems there are plethora of options available. According to some authors, a combination on various approach will give best results. However, more research efforts are needed in order to effectively translate scientific findings into clinical practice.

Corrections made

- please pay attention to the spelling of the abbreviation UTI or CAUTI. On page 8 there should be eg CAUTI's

Correction made

Sincerely,

the authors